# Bio-Inspired Autonomous Navigation and Formation Controller for Differential Mobile Robots

**DOI:** 10.3390/e25040582

**Published:** 2023-03-28

**Authors:** Alejandro Juarez-Lora, Alejandro Rodriguez-Angeles

**Affiliations:** 1Centro de Investigacion en Computacion del Instituto Politecnico Nacional, CIC-IPN, Ciudad de Mexico 07738, Mexico; 2Centro de Investigacion y de Estudios Avanzados del Instituto Politecnico Nacional, Cinvestav-IPN, Ciudad de Mexico 07360, Mexico

**Keywords:** autonomous navigation, multi-agent systems, decentralized control, bio-inspired control, reaction control, formation

## Abstract

This article proposes a decentralized controller for differential mobile robots, providing autonomous navigation and obstacle avoidance by enforcing a formation toward trajectory tracking. The control system relies on dynamic modeling, which integrates evasion forces from obstacles, formation forces, and path-following forces. The resulting control loop can be seen as a dynamic extension of the kinematic model for the differential mobile robot, producing linear and angular velocities fed to the mobile robot’s kinematic model and thus passed to the low-level wheel controller. Using the Lyapunov method, the closed-loop stability is proven for the non-collision case. Experimental and simulated results that support the stability analysis and the performance of the proposed controller are shown.

## 1. Introduction

Autonomous driving on a multi-agent system is a highly researched goal pursued in robotics. *Deliberative* control techniques, which require precise recognition and knowledge of the surroundings and well-defined hierarchical structures, show an efficient performance in controlled environments; see [1], where a broad review of path planning strategies is presented, including environment and robot interaction modeling. In contrast, *reactive* control techniques, based on a robust stimulation–response behavior, yield a favorable execution on unknown environments [2]. The location of the agents (mobile robots) about a global frame of reference, navigation in unknown environments, and the introduction of multiple robots in the same workspace make this a challenging problem to solve. Multiple mobile robots systems present new challenges compared to single vehicle control, such as the heterogeneity of the agents, mixed traffic, cooperative behaviors, collisions among vehicles, static obstacles, and even pedestrians [3,4].

Mobile robots are electro-mechanical devices that can move on the bi-dimensional plane, and this restriction leads to 3 degrees of freedom, which are its location and orientation coordinates. Mobile robots can be classified according to their grade of mobility and directional capabilities. Type (2, 0) or differential robots have two wheels with individual speeds and no directional actuator. Thus, differential mobile robots are underactuated systems subject to non-holonomic constraints, which hinders the design of the navigation controller. Although the navigation control of mobile robots is not new, novel applications and developments are studied, due to the recent technological advances and cost reduction in electronics, as well as more powerful, highly integrated but smaller computing devices [5,6]. In [5], the application of multi-robot systems for harvester assisting purposes is presented such that a cooperative co-working set of robots must follow the farmer to help with weeding, harvesting, crop scouting, etc., rendering a human–machine collaboration system. Other applications for multi-agent systems and social robots navigation, such as delivery robots, warehouses, indoor service robots, surveillance robots, etc., are listed in [7]. An interesting application is presented in [8], where a multi-robot system is proposed for defect detection and location on large surface metal plates; it is pointed out that there are several trade-offs between the performance in terms of defect location accuracy and the number of robots, their navigation capabilities, and ability to keep a formation.

The implementation of multi-agent swarm systems usually involves the usage of several data acquisition devices, such as inertial sensors [9], simultaneous localization and mapping (SLAM) [10], onboard cameras for computer vision [11], and relative radio positioning [12], in order to provide data redundancy and diversity to model environmental conditions properly [13]. There are even biomimetic sensors designed for navigation purposes in multi-robot systems; see [14], where a micro lenses array is integrated to aperture and field diaphragms to emulate an insect compound eye and later is combined to SLAM techniques for the navigation of a set of outdoor light robots. In general, the amount of data processing is directly proportional to the number of agents in the system, increasing power consumption in onboard processing platform solutions. Centralized swarm navigation schemes usually share all their gathered data through all the agents in the system, making them utterly dependent on the communication system’s reliability [15]. To overcome the burden of data required to create a map, in [6], they propose using OpenStreetMaps, which are user-generated maps, publicly available, whose information is combined with lidar-based Naive-Valley-Path generation methodologies to render a local path that is free of obstacle collisions with other vehicles and pedestrians, thus combining deliberative and reactive techniques, providing a complete outdoor autonomous navigation system for unstructured environments. Another application of integrating deliberative and reactive techniques is found in [16], where a navigation controller that uses SLAM, path-planning techniques and exteroceptive sensors is applied to small-scale vessels or surface vehicles.

On the other hand, some species in nature have developed effective ways to achieve evasion, following other members and shaping formation, yielding to behavior models of biological systems, such as schools of fish, flocks of birds, or crowd dynamics, which are examples of reactive control techniques, and can be used for its implementation on robotic systems, more specifically, unicycle differential robots. In these reactive models, the individual actions of each agent are defined toward the fulfillment of a global objective [17,18] so as to have the most number of animals in a reduced space or some species of birds that can make more complex formations toward a leader.

Some advances in the research of crowd dynamics allow, with a set of simple rules, to predict the agents’ behavior. For instance, in [19], potential functions and panel methods are embedded into an algorithm to create a collision-free path for differential agents. The simplicity, ease of implementation, and low computational requirements make this modeling approach effective in robot navigation. However, potential functions may suffer from several problems, such as local minima traps, dead locks between close obstacles, oscillations in the presence of obstacles, and inside narrow passages.

Another kind of solution is obtained from observing the animal’s conduct when moving and interacting with each other, according to its goal. In [20,21], Helbing et al. (2000, 2005) analyzed crowd dynamics, describing a model of displacement behavior; if an individual wants to move to a desired point, it moves in the shortest path with the most comfortable speed; when an obstacle, like a wall or another individual appears, the individual starts evading it as soon as a comfort zone is invaded, i.e., a private space. This zone is preferred not to be invaded, so the individual tries to keep it clear.

There are other investigations, such as [17], which set navigation rules with the purpose of the agents reaching geometric shape formations regarding the position of the other agents in a sort of coordinated behavior. In humans, for example, making a line or a circle implies knowing the location of everybody else in the group such that everyone sets their position on the formation.

More recently, reinforcement learning (RL) and deep reinforcement learning (DRL) techniques have been applied for the navigation of autonomous vehicles; [4,7] present surveys of advances on applying RL and DRL to problems such as obstacle avoidance, indoor navigation, multi-robot navigation, and social navigation, considering heterogeneous fleets, unmanned vehicles, aerial vehicles and ships, and possible interaction between agents and humans. They point out that the results highly depend of the degree of cooperation and shared information between the agents. Particular applications of RL to the case of groups of autonomous vehicles give rise to the so-called multi-agent reinforcement learning (MARL), which is a more distributed framework in which several agents simultaneously learn cooperative or competitive behaviors [22]. This approach is being applied to mixed traffic problems and heterogeneous group of agents.

In this article, a decentralized control strategy is designed, allowing autonomous driving for a multi-agent system, using behavioral models to enable the trajectory tracking and collision avoidance of dynamic and static obstacles, based on the relative approaching velocity. A desired formation is simultaneously achieved by establishing some navigation restrictions, in this case, and without loss of generality, a circular formation. The objective of the research is then to use these models and to adapt them in a control strategy which allows a decentralized, autonomous navigation system.

In Section 2, the specifications of the control proposal are described, while in Section 3, the Lyapunov’s stability test is applied on the closed-loop system, for the non-collision case, to obtain stability conditions to be satisfied by the control gains. In Section 4, the development of an experimental platform is shown. Section 5 compares the simulation and experimental results. Section 6 gives the conclusions of this work.

## 2. Design of the Autonomous Navigation Control Law

A differential mobile robot type (2, 0) has the kinematic model described in (Equation 1), where $\left[x_{i},y_{i}\right]$ are the coordinates of the rotation center regarding a fixed frame of reference, and $\theta_{i}$ is its orientation, as shown in Figure 1. Subscript *i* identifies the *i*-th robot in a multi-agent system. For this model, the control inputs are the translational velocity $V_{i}$ and rotational velocity $W_{i}$, [23].
(1)$$\begin{array}{c} {\dot{x}}_{i}=V_{i}cos\theta_{i}\\ {\dot{y}}_{i}=V_{i}sin\theta_{i}\\ {\dot{\theta }}_{i}=W_{i} \end{array}$$

As mentioned in the introduction, the proposed controller integrates a dynamic model based on reactive forces into the kinematic model, controlled by the translational velocity $V_{i}$ and rotational velocity $W_{i}$. The integration can be depicted in a nested control scheme formed by an external and internal controller. The *external* control loop contains the bio-inspired navigation model, which is a driven force dynamic model, whose output is processed by the *internal* control loop that corresponds to the kinematic model and is driven by the translational $V_{i}$ and rotational velocity $W_{i}$. This interconnection is shown in Figure 2. In the following, the integrated controller is explained in detail.

### 2.1. External Control Loop

This control loop consists of the bio-inspired algorithm, which is built in a series of modifications of Helbing’s crowd dynamics model, presented in [20,21], which portrays the behavior of an individual of mass $m_{i}$ and velocity $v_{i}=\left[v_{x_{i}},v_{y_{i}}\right]$ given by a set of socio-psychological and physical rules. The individual’s position $p_{i}=\left[x_{i},y_{i}\right]$ moves toward a reference point or goal position $p_{d_{i}}=\left[x_{d_{i}},y_{d_{i}}\right]$. The desired individual’s velocity, denoted as the vector $v_{i}^{0}$, is reached in a characteristic transition time $\tau_{i}$, and points towards a direction given by the position error vector $e_{i}=\left[x_{d_{i}}-x_{i},y_{d_{i}}-y_{i}\right]$, therefore moving in a straight line to the goal position. At the same time, it intends to avoid obstacles and other individuals which invade its comfort zone $r_{i}$. Evasion forces for dynamic and static obstacles are denoted as $f_{ij}$ and $f_{iw}$, respectively. This dynamic force model, given by (Equation 2), is then composed by attracting forces related to the goal position and repulsive forces based on the distance to obstacles, yielding a reactive control.
(2)$$\begin{array}{c} m_{i}\left[\begin{array}{c} {\dot{v}}_{x_{i}}\\ {\dot{v}}_{y_{i}} \end{array}\right] \end{array}=\frac{m_{i}}{\tau_{i}}\left(v_{i}^{0}e_{i}-v_{i}\right)+\sum f_{ij}+\sum f_{iw}$$

The obstacle repulsive forces $f_{ij}$ and $f_{iw}$ are modeled by (Equation 3) and (Equation 4), and correspond to the evasion of dynamic or static obstacles, respectively, considering as dynamic obstacles other agents that get into the comfort zone of the *i*-th robot. These repulsive forces are based on relative distances that onboard sensors can quickly obtain:(3)$$\begin{array}{c} f_{ij}=k_{i}g\left(r_{ij}-d_{ij}\right)n_{ij}-\kappa_{i}g\left(r_{ij}-d_{ij}\right)\Delta v_{ji}^{t}t_{ij} \end{array}$$
(4)$$\begin{array}{c} f_{iw}=k_{i}g\left(r_{i}-d_{iw}\right)n_{ij}-\kappa_{i}g\left(r_{i}-d_{iw}\right)\left(v_{i}\cdot t_{iw}\right)t_{iw} \end{array}$$

This model has terms as the distance between the mass center of agent *i* and *j* denoted by $d_{ij}=\left|\right|p_{i}-p_{j}\left|\right|$, the normalized vector of direction $n_{ij}=\left[n_{ij_{x}},n_{ij_{y}}\right]=\left(p_{i}-p_{j}\right)/d_{ij}$, which points from the agent *j* toward agent *i*, the tangential vector $t_{ij}=\left[-n_{ij_{y}},n_{ij_{x}}\right]$, the term *g*, which determines if the comfort zone of agent *i* interacts with the agent *j* comfort zone, i.e., g=1 if $r_{ij}>d_{ij}$ (where $r_{ij}=r_{i}+r_{j}$, with $r_{i},r_{j}$ the radii of the comfort zones) and g=0 otherwise. The difference in translational velocity vectors between the *i* and *j* agents represents a relative approaching velocity, and it is denoted by $\Delta v_{ji}^{t}=\left(v_{j}-v_{i}\right)\cdot t_{ij}$. The magnitude of the positive gains *k* and κ determines the influence of the normal and tangential components of acceleration, which permits modulating the response of acceleration and intensity of twist to avoid collisions. The repulsion forces against static obstacles $f_{iw}$ comes as setting $v_{j}=0$ in (Equation 3), with $d_{iw}$ being the distance between the *i* agent and the obstacle once it gets into the agent’s comfort zone.

The dynamic model (Equation 2) is used for designing a dynamic control law at acceleration level driven by forces, whose states are fed to the kinematic model of the mobile robots as depicted in Figure 2. For this purpose, consider that the mass $m_{i}$ is unitary, and any obstacle, either dynamic or static, can be treated with the same evasion function, taking into consideration only its relative approaching velocity. Furthermore, friction forces are let out of the model as well as slippery forces in order to simplify the control design; nevertheless, taking into consideration such forces would improve the trajectory tracking performance. With all of the above considerations and based on the control objectives, the control law given by (Equation 5) is proposed:(5)$$\begin{array}{c} {\dot{v}}_{i}=\left[\begin{array}{c} {\dot{v}}_{x_{i}}\\ {\dot{v}}_{y_{i}} \end{array}\right]=k_{p}e_{i}+k_{d}{\dot{e}}_{i}+\sum f_{ij}+f_{ic} \end{array}$$
where in order to achieve trajectory tracking, a PD controller is included with tuning gains $k_{p},k_{d}$ and fed by the position error (Equation 6); meanwhile, the total repulsive forces are the vector sum of all avoiding collision forces $f_{ij}$, related to dynamic and static obstacles that are inside the comfort zone, taking into account that the relative approaching velocity for dynamic obstacles (other vehicles) is given by $\Delta v_{ji}^{t}=\left(v_{j}-v_{i}\right)\cdot t_{ij}$, while for static obstacles, the approaching velocity corresponds only to the velocity of the agent itself as shown on the last terms of Equations (Equation 3) and (Equation 4). The term $f_{ic}$ allows enforcing a desired geometric formation of the multi-agent system, then the position of each agent in the formation must be congruent with satisfying the desired trajectory of the agent. Otherwise, a conflict would arise, and none of the goals, nor trajectory tracking, nor formation would be satisfied. By definition of $f_{ij}$, several simple geometric shape formations may be generated [17], such as straight lines, arrow shapes, circles, etc.:(6)$$\begin{array}{c} e_{i}=\left[\begin{array}{c} e_{x_{i}}\\ e_{y_{i}} \end{array}\right]=\left[\begin{array}{c} x_{d_{i}}-x_{i}\\ y_{d_{i}}-y_{i} \end{array}\right] \end{array}$$

Based on the Cucker–Smale model modifications presented in [18], the function $f_{ic}$ is defined as in (Equation 7) that corresponds to a circular formation:(7)$$f_{ic}=\gamma \left(1-\frac{R}{\left|\left|d_{ic}\right|\right|}\right)d_{ic}$$

The function $f_{ic}$ pretends to form a circle of radio *R* with the *N* agents of the system, regarding the geometric center $\left(\bar{x},\bar{y}\right)$, which is given by the average position of the whole system, (Equation 8). The distribution of each agent on the circle formation takes into account the desired trajectory for each agent, as would be shown at the presented results:(8)$$\bar{x}=\frac{1}{N}\sum_{i=1}^{N}x_{i},\;\bar{y}=\frac{1}{N}\sum_{i=1}^{N}y_{i}$$

In function (Equation 7), $d_{ic}={\left[\bar{x}-x_{i},\bar{y}-y_{i}\right]}^{T}$ is the vector distance from agent *i* toward the geometric center, and γ is a positive gain for tuning. By evaluating the control proposed at (Equation 5) and integrating it, the values $\left({\dot{v}}_{x_{i}},{\dot{v}}_{y_{i}},v_{x_{i}},v_{y_{i}}\right)$ are obtained.

### 2.2. Internal Control Loop

This control loop transforms the velocities and accelerations obtained by the *external control loop* into the linear and angular velocities $V_{i}$ and $W_{i}$ needed to drive the vehicle. To achieve this, it is considered the work at [24], obtaining the relations given in (Equation 9) and (Equation 10):(9)$$\begin{array}{c} V_{i}=k_{v}\left[v_{x_{i}}cos\left(\theta_{i}\right)+v_{y_{i}}sin\left(\theta_{i}\right)\right] \end{array}$$
(10)$$\begin{array}{c} W_{i}=k_{a}\left[\frac{{\dot{v}}_{y_{i}}v_{x_{i}}-{\dot{v}}_{x_{i}}v_{y_{i}}}{v_{x_{i}}^{2}+v_{y_{i}}^{2}+\epsilon }sin_{c}\left(e_{\theta_{i}}\right)-k_{t}e_{\theta_{i}}\right] \end{array}$$
where $k_{v}$ is a gain which modulates the intensity of the control signal $V_{i}$. The constant ϵ≈0 is added in order to avoid singularities when $v_{x_{i}},v_{y_{i}}$ are zero. The function $sin_{c}\left(\cdot \right)$ is considered the *cardinal sine*, and it is defined in (Equation 11):(11)$$sin_{c}\left(\alpha \right)=\left\{\begin{array}{ccc} \frac{sin(\alpha )}{\alpha } & if & \alpha \neq 0\\ 1 & if & \alpha =0 \end{array}\right.$$

Since it is intended to achieve a desired orientation, a proportional control is added with $e_{\theta_{i}}=\left(\theta_{i}-\theta_{d_{i}}\right)$ as the orientation error regarding an angle of reference $\theta_{d_{i}}$. The value of $k_{a},\;k_{t}$ modulatesthe control actions given by the orientation control.

## 3. Stability Analysis for the Non-Collision Case

The avoidance collision force term $f_{ij}$ represents repulsive forces for dynamic and static obstacles, depending on the relative approaching velocity. Therefore, such forces depend on each possible scenario that the agent may encounter, and thus the stability analysis is carried out for the free collision case. From the simulation and experimental tests, it is concluded that the proper tuning of repulsive forces will not affect convergence to the desired position as far as there is not an obstacle inside the comfort zone, generating a conflict for the agent being in its desired position.

Using Lyapunov’s stability test, setting $f_{ij}=0$ for the collision-free case and taking into consideration the values of interest $\left(x_{i},y_{i},v_{x_{i}},v_{y_{i}},\theta_{i}\right)$ and their desired references $\left(x_{d_{i}},y_{d_{i}},v_{xd_{i}},v_{yd_{i}},\theta_{d_{i}}\right)$, the next state variables are defined:(12)$$\begin{array}{cc} z_{1_{i}}=e_{x_{i}}=x_{d_{i}}-x_{i} & z_{2_{i}}=e_{y_{i}}=y_{d_{i}}-y_{i}\\ z_{3_{i}}=e_{v_{x_{i}}}=v_{x_{d_{i}}}-v_{x_{i}} & z_{4_{i}}=e_{v_{y_{i}}}=v_{y_{d_{i}}}-v_{y_{i}}\\ z_{5_{i}}=e_{\theta_{i}}=\theta_{i}-\theta_{d_{i}} \end{array}$$

The closed loop for the *i*-th agent is given by
(13)$$\begin{array}{c} {\dot{z}}_{1_{i}}=v_{x_{d_{i}}}sin^{2}\theta_{i}+\left(z_{4_{i}}-v_{y_{d_{i}}}\right)sin\left(\theta_{i}\right)cos\left(\theta_{i}\right)+z_{3_{i}}cos^{2}\left(\theta_{i}\right)\\ {\dot{z}}_{2_{i}}=v_{y_{d_{i}}}cos^{2}\left(\theta_{i}\right)+\left(z_{3_{i}}-v_{x_{d_{i}}}\right)sin\left(\theta_{i}\right)cos\left(\theta_{i}\right)+z_{4_{i}}sin^{2}\left(\theta_{i}\right)\\ {\dot{z}}_{3_{i}}={\dot{v}}_{x_{d_{i}}}-k_{p}z_{1_{i}}-k_{d}z_{3_{i}}-\gamma \beta_{i_{x}}\\ {\dot{z}}_{4_{i}}={\dot{v}}_{y_{d_{i}}}-k_{p}z_{2_{i}}-k_{d}z_{4_{i}}-\gamma \beta_{i_{y}}\\ {\dot{z}}_{5_{i}}=-{\dot{\theta }}_{d_{i}}-k_{a}k_{t}z_{5_{i}}+k_{a}\left[\frac{\left({\dot{v}}_{y_{d_{i}}}-{\dot{z}}_{4_{i}}\right)\left(v_{x_{d_{i}}}-z_{3_{i}}\right)-\left(v_{y_{d_{i}}}-z_{4_{i}}\right)\left({\dot{v}}_{x_{d_{i}}}-{\dot{z}}_{3_{i}}\right)}{{\left(v_{x_{d_{i}}}-z_{3_{i}}\right)}^{2}+{\left(v_{y_{d_{i}}}-z_{4_{i}}\right)}^{2}+\epsilon }\left(\frac{sin(z_{5_{i}})}{z_{5_{i}}}\right)\right] \end{array}$$

With ${\dot{\theta }}_{d_{i}}=W_{d_{i}}$ as angular velocity reference and
(14)$$\beta_{i}=\left[\begin{array}{c} \beta_{i_{x}}\\ \beta_{i_{y}} \end{array}\right]=\left(1-\frac{R}{\sqrt{{\left(\bar{x}-x_{d_{i}}+z_{1_{i}}\right)}^{2}+{\left(\bar{y}-y_{d_{i}}+z_{2_{i}}\right)}^{2}}}\right)\left[\begin{array}{c} \bar{x}-x_{d_{i}}+z_{1_{i}}\\ \bar{y}-y_{d_{i}}+z_{2_{i}} \end{array}\right]$$

Using the Lyapunov candidate function $V_{i}\left(z_{i}\right)=\frac{1}{2}z_{i}^{T}z_{i}\geq 0$, the time derivate of $V_{i}\left(z_{i}\right)$ is
(15)$$\begin{array}{c} {\dot{V}}_{i}\left(z_{i}\right)=\left[cos^{2}\theta_{i}-k_{p}\right]z_{1_{i}}z_{3_{i}}+\left[sin^{2}\theta_{i}-k_{p}\right]z_{2_{i}}z_{4_{i}}\\ +\left[sin\theta_{i}cos\theta_{i}+\frac{k_{a}k_{p}sin\left(z_{5_{i}}\right)}{{\left(v_{x_{d_{i}}}-z_{3_{i}}\right)}^{2}+{\left(v_{y_{d_{i}}}-z_{4_{i}}\right)}^{2}+\epsilon }\right]z_{1_{i}}z_{4_{i}}\\ +\left[sin\theta_{i}cos\theta_{i}-\frac{k_{a}k_{p}sin\left(z_{5_{i}}\right)}{{\left(v_{x_{d_{i}}}-z_{3_{i}}\right)}^{2}+{\left(v_{y_{d_{i}}}-z_{4_{i}}\right)}^{2}+\epsilon }\right]z_{2_{i}}z_{3_{i}}\\ -\left[sin\theta_{i}cos\theta_{i}+\frac{k_{a}k_{p}sin\left(z_{5_{i}}\right)}{{\left(v_{x_{d_{i}}}-z_{3_{i}}\right)}^{2}+{\left(v_{y_{d_{i}}}-z_{4_{i}}\right)}^{2}+\epsilon }\right]z_{1}v_{y_{d_{i}}}\\ -\left[sin\theta_{i}cos\theta_{i}+\frac{k_{a}k_{p}sin\left(z_{5_{i}}\right)}{{\left(v_{x_{d_{i}}}-z_{3_{i}}\right)}^{2}+{\left(v_{y_{d_{i}}}-z_{4_{i}}\right)}^{2}+\epsilon }\right]z_{1}v_{x_{d_{i}}}\\ +\left[z_{1_{i}}sin\theta_{i}+\gamma \beta_{i_{y}}\left(\frac{k_{a}k_{p}sin\left(z_{5_{i}}\right)}{{\left(v_{x_{d_{i}}}-z_{3_{i}}\right)}^{2}+{\left(v_{y_{d_{i}}}-z_{4_{i}}\right)}^{2}+\epsilon }\right)\right]v_{x_{d_{i}}}\\ +\left[z_{2_{i}}cos\theta_{i}-\gamma \beta_{i_{x}}\left(\frac{k_{a}k_{p}sin\left(z_{5_{i}}\right)}{{\left(v_{x_{d_{i}}}-z_{3_{i}}\right)}^{2}+{\left(v_{y_{d_{i}}}-z_{4_{i}}\right)}^{2}+\epsilon }\right)\right]v_{y_{d_{i}}}\\ -k_{d}\left[z_{3_{i}}^{2}+z_{4_{i}}^{2}\right]-k_{a}k_{t}z_{5_{i}}^{2}-z_{5_{i}}W_{d_{i}} \end{array}$$

In order to prove ${\dot{V}}_{i}\left(z_{i}\right)<0$, we can upper bound it considering the algebraic properties $(a^{2}+b^{2})\geq 0$ and $\frac{1}{2}\left(a^{2}+b^{2}\right)\geq -ab$. Furthermore, it is worth mentioning that the motors produce a maximal velocity and acceleration, which are positive and bounded, regardless of its spin direction. So, it results in $|v_{x_{d_{i}}}|<v_{xi_{max}}$, $|v_{y_{d_{i}}}|<v_{yi_{max}}$ and $|{\dot{v}}_{x_{d_{i}}}|\leq {\dot{v}}_{xi_{max}}$, $|{\dot{v}}_{y_{d_{i}}}|\leq {\dot{v}}_{yi_{max}}$. Replacing the bounds of the trigonometrical functions in (Equation 15) given by $-0.2\leq cos^{2}\theta_{i}+sin\theta_{i}cos\theta_{i}\leq 1.2$, $-0.2\leq sin^{2}\theta_{i}+sin\theta_{i}cos\theta_{i}\leq 1.2$, $|sin(z_{5_{i}})|\leq 1$ and considering the worst-case operation scenario, the function can be written as
(16)$$\begin{array}{c} \dot{V}\left(z_{i}\right)\leq \frac{1}{2}\left[k_{p}\left(1-\frac{k_{a}}{\epsilon }\right)+0.2\right]z_{1_{i}}^{2}+\frac{1}{2}\left[k_{p}\left(1+\frac{k_{a}}{\epsilon }\right)+0.2\right]z_{2_{i}}^{2}\\ +\frac{1}{2}\left[k_{p}\left(1+\frac{k_{a}}{\epsilon }\right)+0.2\right]z_{3_{i}}^{2}+\frac{1}{2}\left[k_{p}\left(1-\frac{k_{a}}{\epsilon }\right)+0.2\right]z_{4_{i}}^{2}\\ +\left[v_{x_{i_{max}}}+\left(\frac{1}{2}-k_{p}\frac{k_{a}}{\epsilon }\right)v_{y_{i_{max}}}\right]z_{1_{i}}+\left[v_{y_{i_{max}}}+\left(\frac{1}{2}-k_{p}\frac{k_{a}}{\epsilon }\right)v_{x_{i_{max}}}\right]z_{2_{i}}\\ +\left[{\dot{v}}_{x_{i_{max}}}-\gamma \beta_{i_{x}}-\left(\gamma \beta_{i_{y}}+k_{d}v_{y_{i_{max}}}\right)\frac{k_{a}}{\epsilon }\right]z_{3_{i}}\\ +\left[{\dot{v}}_{y_{i_{max}}}-\gamma \beta_{i_{y}}+\left(\gamma \beta_{i_{x}}+k_{d}v_{x_{i_{max}}}\right)\frac{k_{a}}{\epsilon }\right]z_{4_{i}}\\ +\frac{k_{a}}{\epsilon }\left[\gamma \beta_{i_{y}}v_{x_{i_{max}}}-\gamma \beta_{i_{x}}v_{y_{i_{max}}}\right]-k_{d}\left[z_{3_{i}}^{2}+z_{4_{i}}^{2}\right]-k_{a}k_{t}z_{5_{i}}^{2}-z_{5_{i}}W_{d_{i}} \end{array}$$

According to (Equation 7), the best-case scenario is when agent *i* is in a circular formation, i.e., $\left|\right|d_{ic}\left|\right|=R$, yielding to $\left|\right|\beta_{i}\left|\right|=0$. The worst-case scenario is $\left|\right|d_{ic}\left|\right|>R$, which means the agent *i* is outside the circumference and $\left|\right|\beta_{i}\left|\right|\neq 0$. These considerations lead to the next list of conditions, which accomplish ${\dot{V}}_{i}\left(0\right)<0$:(17)$$\begin{array}{cc} 1. & k_{p}>0,k_{d}>0,k_{a}>0,k_{t}>0,\epsilon >0,\gamma >0;\\ 2. & k_{p}>\frac{0.2}{\left(1-\frac{k_{a}}{\epsilon }\right)};\\ 3. & k_{p}>\frac{\epsilon }{k_{a}}\left(\frac{v_{x_{i_{max}}}}{v_{y_{i_{max}}}}+\frac{1}{2}\right);\\ 4. & k_{d}>\frac{{\dot{v}}_{x_{i_{max}}}}{v_{y_{i_{max}}}}\frac{\epsilon }{k_{a}};\\ 5. & \gamma >\frac{\frac{k_{a}}{\epsilon }k_{d}v_{y_{i}max}-{\dot{v}}_{x_{i_{max}}}}{1+\frac{k_{a}}{\epsilon }},\;\gamma <\frac{{\dot{v}}_{y_{i_{max}}}+\frac{k_{a}}{\epsilon }k_{d}v_{x_{i_{max}}}}{1+\frac{k_{a}}{\epsilon }}. \end{array}$$

Under the conditions given in (Equation 17), the closed-loop system is asymptotically stable only when $\left|\right|\beta_{i}\left|\right|=0$ as $V_{i}\left(0\right)=0$, $V_{i}\left(z_{i}\right)>0$, ${\dot{V}}_{i}\left(0\right)=0$ and ${\dot{V}}_{i}\left(z_{i}\right)<0$. Nevertheless, when $\left|\right|\beta_{i}\left|\right|\neq 0$, there are unvanishing terms, which makes ${\dot{V}}_{i}\left(0\right)\neq 0$ and only retrieves practical stability, known as uniformly ultimately bounded (UUB) stability.

To perform stability analysis considering possible collisions with dynamic and static obstacles, each different reaction force for each possible situation during the navigation of the agent should be properly modeled, which is a cumbersome task. Nevertheless, simulation and experimental tests showed that tuning the reaction forces for dynamic obstacles would also work for static obstacles since for the last ones, the relative approaching velocity is smaller than that of dynamic obstacles, thus requiring less aggressive evasion actions. Then, k,κ in expression (Equation 3) are tuned so the repulsive forces can produce a quick response to avoid collisions with dynamic obstacles; the same tuning is used for static obstacles, while maintaining the system’s stability, assuming bounded and differentiable perturbations.

## 4. Experimental Platform Considerations

In order to prove the proposed control law, a MATLAB^®^ [25] simulation and a physical implementation are prepared. During experimentation, four Turtlebot3 Burger^®^ and one Turtlebot3 Waffle Pi^®^ are used, giving a total of 5 agents, individually controlled using ROS in an onboard Raspberry Pi^®^, where the external control law is deployed, computing the control inputs $V_{i}$, $W_{i}$ for each agent, delivered to the onboard OpenCR^®^ power stage card through serial protocol. The description of the hardware and the programmed nodes in ROS is available in Figure 3.

The initial position of each robot, on the same inertial global frame, is given, so each agent, using odometry, can compute its position on each given time. The used algorithms to obtain parameters needed in the control loop are described next.

### 4.1. Trajectory Reference Generation

The function in (Equation 18) retrieves a lemniscate figure, and it can be computed to obtain the reference position, velocity and acceleration of each agent $x_{d_{i}}\left(t\right),y_{d_{i}}\left(t\right)$: (18)$$\begin{array}{cc} x_{d_{i}}\left(t\right)=acos\left(k_{s}t\right)+Rcos\left(\frac{360(i)}{N}\right) & y_{d_{i}}\left(t\right)=bsin\left(2k_{s}t\right)+Rsin\left(\frac{360(i)}{N}\right)\\ {\dot{x}}_{d_{i}}\left(t\right)=-a\cdot k_{s}\cdot sin\left(k_{s}t\right) & {\dot{y}}_{d_{i}}=2b\cdot k_{s}\cdot cos\left(2k_{s}t\right)\\ {\ddot{x}}_{d_{i}}\left(t\right)=-a\cdot k_{s}^{2}\cdot cos\left(k_{s}t\right) & {\ddot{y}}_{d_{i}}\left(t\right)=-4bk_{s}^{2}\cdot sin\left(k_{s}t\right) \end{array}$$

Here, a,b are the length and width of the lemniscate, *R* is the formation circle radius, *N* is the number of agents, the sub-index *i* refers to the *i*-th agent, and $k_{s}$ is the temporary exchange rate.

The trajectory reference has a duration *d*, where if t<d, the function and its time derivatives are computed. If t>d, the last computed value on t=d is used in order to obtain the final position coordinates of the trajectory, with reference velocity equal to zero.

Once the references are obtained, the expressions in (Equation 19)–(Equation 21) are used to obtain $\theta_{d_{i}},V_{d_{i}},W_{d_{i}}$:(19)$$\begin{array}{c} \theta_{d_{i}}=tan^{-1}\left(\frac{{\dot{x}}_{d_{i}}}{{\dot{y}}_{d_{i}}}\right) \end{array}$$
(20)$$\begin{array}{c} V_{d_{i}}=\sqrt{{\dot{x}}_{d_{i}}^{2}+{\dot{y}}_{d_{i}}^{2}} \end{array}$$
(21)$$\begin{array}{c} W_{d_{i}}=\frac{{\dot{x}}_{d_{i}}{\ddot{y}}_{d_{i}}-{\dot{y}}_{d_{i}}{\ddot{x}}_{d_{i}}}{v_{d_{i}}^{2}+\epsilon } \end{array}$$

### 4.2. Perception

The agent’s comfort zone consists of a continuous area around the agent. This area can be set as several shapes, which matches the surrounding obstacles. For simplicity, for the analysis performed in this work, it is chosen to be circular, assuming that potential obstacles in the surrounding space of the reactive behavior of the robot should conform to the circular-shaped area, as shown in a wide variety of cases in nature, e.g., in agriculture, as the robotic solution proposed in [5].

To measure the distance $d_{ij}$ from the agent *i* to the agent *j*, and the angle $\theta_{ij}$ about the agent *i*’s translational axis, where agent *j* is located, it is necessary to implement the comfort zone properly. An infrared sensor emits a beam of light, which, when reflected on an object’s surface, enables measurement of the distance from where it is located. Setting multiple infrared sensors around the agent is impractical due to the poor retrieved resolution due to the number of beams on the circumference, which is limited to the available space. This can be observed in Figure 4a, where an obstacle is not detected despite being in the comfort zone. A lidar system allows increasing the resolution, spinning an infrared beam to cover all the circumference. The LDS-01, used in this work, has a 1/360 resolution, scilicet, emits a beam each 0.0174 [rad] [26]. This is a 360-beam configuration around the agent and improves the environment perception, as many beams are now reflected in a single object.

In this event, the shortest registered distance is set as $d_{ij}$, while $\theta_{ij}$ is set as the average angle of all the triggered beam lights. When there are multiple beams activated, but these are not consecutive, it is considered to be multiple-obstacle detection, as shown in Figure 4b.

### 4.3. Position and Velocity Estimation of the Other Agents

For computing collision avoidance function $f_{ij}$ given by (Equation 3) and used in (Equation 5), the position and velocity of the *j*-th agent are needed. These are denoted as $p_{j}={\left[x_{j},y_{j}\right]}^{T}$ and $v_{j}={\left[{\dot{x}}_{j},{\dot{y}}_{j}\right]}^{T}={\left[v_{x_{j}},v_{y_{j}}\right]}^{T}$, and in order to compute them, the decentralized estimation algorithm starts from these suppositions:Each agent knows its own position regarding a global reference frame.All the obstacles detected in a radio detection $R_{d}$ are considered agents.The comfort zone of the agent *j* is assumed to be the same radius as agent *i*.

The next algorithm is executed:Agent *i* determines with its sensors the vector to the agent *j*, which is composed by the distance $d_{ij}$, and the angle from the global *x* axis frame, denoted as $\rho_{ij}$.To obtain the $\rho_{ij}$ angle, the orientation of the agent *i* in its inertial frame $\theta_{i}$ and the angle of the agent *j* detected by the sensor, denoted as $\theta_{ij}$, are needed.While $\theta_{i}$ increases, the value of $\theta_{ij}$ decreases, as shown in Figure 5a. This yields the value $\rho_{ij}$ given by
(22)$$\begin{array}{c} \rho_{ij}=\theta_{i}+\theta_{ij} \end{array}$$Agent *j*’s position on the global reference frame is obtained by
(23)$$x_{j}=x_{i}+d_{ij}sin\left(\rho_{ij}\right),\;y_{j}=y_{i}+d_{ij}cos\left(\rho_{ij}\right)$$The geometric center is computed by
(24)$$\bar{x}=\frac{1}{N_{i}}\sum_{i=1}^{N_{i}}x_{i},\;\bar{y}=\frac{1}{N_{i}}\sum_{i=1}^{N_{i}}y_{i}$$
where $N_{i}$ is the number of agents detected by the *i*-th agent. The fact that $N_{i}\neq N$ is possible must be remarked, due to these reasons:If the agent *j* is out of the sensor coverage area of the agent *i*;If agent *k* is behind the agent *j*, agent *i* will not be detected.


*Note: This only applies for the experimentation platform. In simulation, all variables are available, which disables the scenario depicted in Figure 5b.*


Finally, to determine the velocity of agent *j*, the distance $d_{ij}$ is compared with the distance on the previous execution step, denoted as $d_{ij_{\left[t-1\right]}}$. The expression (Equation 25) obtains the velocity of agent *j*:(25)$$\begin{array}{c} v_{j}=\frac{d_{ij}-d_{ij_{\left[t-1\right]}}}{t-t_{-1}} \end{array}$$

To obtain the velocity on its axis components and to evaluate the tangential velocity $\Delta v_{ji}^{t}$, the next functions are evaluated:(26)$$v_{x_{j}}=v_{j}cos\left(\rho_{ij}\right),\;v_{y_{j}}=v_{j}sin\left(\rho_{ij}\right)$$

## 5. Simulation and Experimental Results

At simulation, comparison studies for the case of trajectory tracking and collision avoidance were carried out, considering other techniques proposed in the literature, such as potential repulsive fields, and the geometric obstacle avoidance control method (GOACM), obtaining similar performance; however, there is not a completely similar strategy that integrates formation, trajectory tracking, and collision avoidance of dynamic and static obstacles, for which full comparison of our proposed strategy could be done. Then, for the sake of space and considering that such comparison studies do not represent a fair layout, such results are not presented here.

The experimental configuration consists of a set of N=5 agents, which has to follow a lemniscate trajectory given by (Equation 18), with the parameters on Table 1 and an execution period of t=100 [s]. Considering the stability conditions given by (Equation 17), the upper bound for the control actions is chosen according to the specification of the TurtleBot 3. This is $V_{max}=0.22$ [m/s] and $W_{max}=2.84$ [rad/s], [26], and the tuning of the control gains is shown in Table 2. Lastly, the initial conditions of each agent are shown in Table 3.

Simulation and experimental results are presented under the same conditions of desired trajectory and tuning gains for comparison purposes. It is essential to emphasize that simulations are an ideal case. At the same time, during experiments, odometry and lidar measurements are used to determine each agent’s location and distance to possible obstacle collision. For the sake of space, only results that involve trajectory tracking, a desired circular formation and possible collisions between agents, i.e., dynamic obstacles, are presented. Since the proposed avoidance collision strategy is based on the relative approaching velocity, it is clear that more aggressive evasion actions are required for dynamic obstacles than for static ones because for dynamic obstacles the relative approaching velocity is higher than that of the proper agent, while for a static obstacle, the approaching velocity corresponds to that of the agent. Collision avoidance among agents is present at the transient, between 0 and 8 s, both in the simulation and experimental results because the initial position of the agents imply possible collisions while trying to get into their assigned position at the formation. Nevertheless, several cases considering static obstacles were tested both by simulation and experiments, showing good obstacle avoidance.

First, the simulation results are shown. Figure 6a shows the geometric center position of the whole system, compared with the ideal group formation reference. A better convergence is exhibited in sections where the curvature radio is prominent. Figure 6b shows the correspondent convergence error, and it can be deduced that for t>85 [s], the geometric center position converges to a standstill position. All the agents converge to the same external and internal control loop action, as shown in Figure 7a,b, thus moving in a synchronized way, while tracking the desired lemniscate trajectory.

As for the experimental setup, technical difficulties and limitations are evident in Figure 8 and Figure 9, where noise and abrupt changes are present. This is because of the lidar measurements and poor odometry location.

Nevertheless, the group tries to follow the lemniscate trajectory reference while generating the circular formation pattern as exhibited in Figure 10a. This can also be noted in the tracking errors in Figure 10b; for t>85 [s], the desired trajectory stops, but since there are position errors of each agent concerning the circular formation, the agent keeps moving to fit into the circumference, but at the end, the circular formation is achieved as shown in the experiment snapshot of Figure 11.

## 6. Conclusions

The proposed control law, which aims to generate a desired formation during path tracking, shows satisfactory performance in simulation and acceptable performance in the experimental setup, taking into account the technical limitations of the experimental platform. Better acquisition of the environment data, achieving recognition between the agents in the system, and other localization mechanisms, such as sensor data fusion and filtering, would be reflected in an improvement of the controller behavior.

The proposed controller can be seen as a dynamic extension of the standard kinematic control because of the incorporation of the force-driven model, which allows obstacle avoidance, trajectory tracking, and enforcing the formation. This dynamic force model can be further modified to include some other goals related to the synchronization of the agents, enclosing, escorting, etc. The proposed controller is decentralized and highly relies on the perception capacities of each agent, but it could be easily implemented based on communication systems such that each agent shares its location and event information of detected obstacles. The overall multi-agent systems would enhance its performance and possible applications, such as harvesting, bodyguard formation, terrain coverage and a possible extension into flying vehicles.

## Figures and Tables

**Figure 1 entropy-25-00582-f001:**
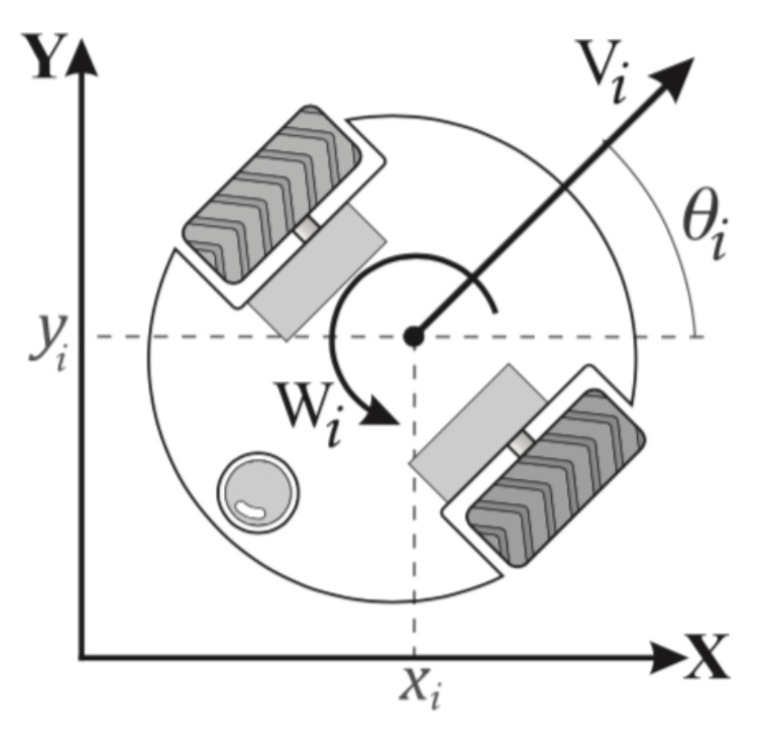
Type (2, 0) Mobile robot.

**Figure 2 entropy-25-00582-f002:**
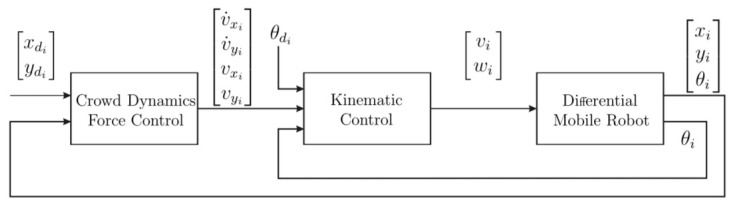
Scheme of the integrated control model.

**Figure 3 entropy-25-00582-f003:**
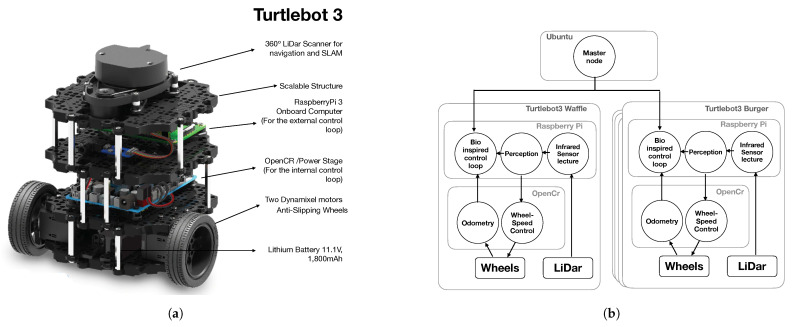
Description of the experimental platform. (**a**) Structure of the used differential re-configurable robot Turtlebot3, (**b**) coded nodes using ROS for the deployment of the internal and external control loops. Each rectangle represents a device, while each circle represents a coded node in ROS.

**Figure 4 entropy-25-00582-f004:**
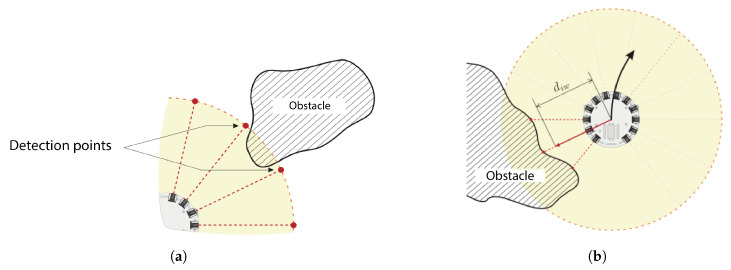
Detection scenarios regarding sensing. (**a**) Detection system issue, (**b**) multiple consecutive beam lights triggered.

**Figure 5 entropy-25-00582-f005:**
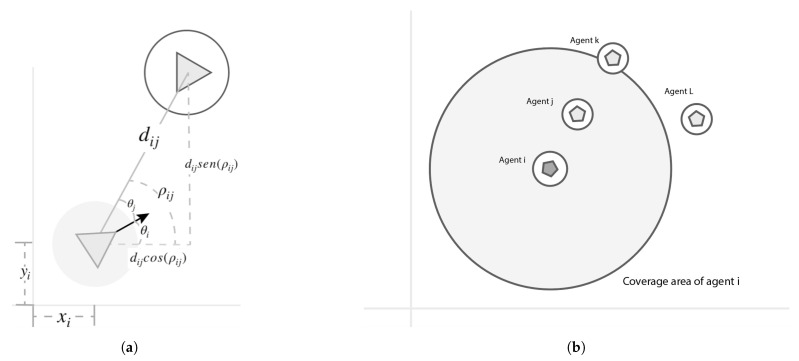
Detection scenarios regarding positioning. (**a**) Position estimation scheme. (**b**) Agent *i* cannot detect the agent *k*, as it is behind agent *j*. Agent *L* cannot be detected because it is outside the agent *i* coverage area.

**Figure 6 entropy-25-00582-f006:**
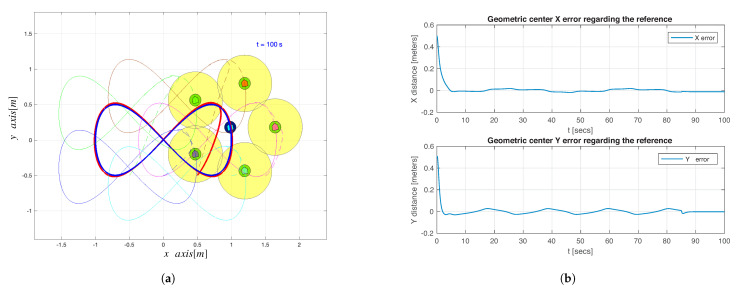
Detection scenarios regarding positioning. (**a**) Geometric center position (Simulation), (**b**) trajectory tracking error of the geometric center (Simulation).

**Figure 7 entropy-25-00582-f007:**
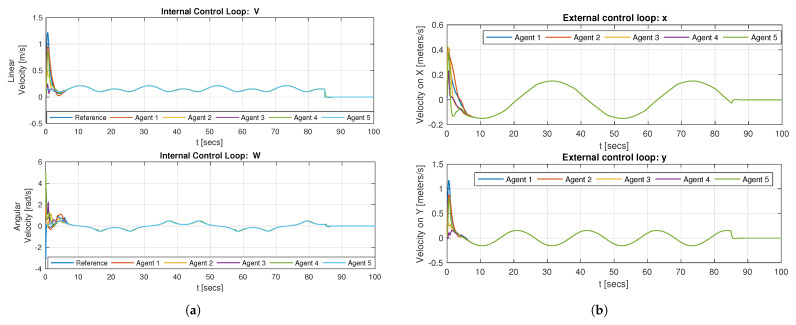
Control signals in simulation setup. (**a**) Internal control loop variables (Simulation), (**b**) external control loop variables (Simulation).

**Figure 8 entropy-25-00582-f008:**
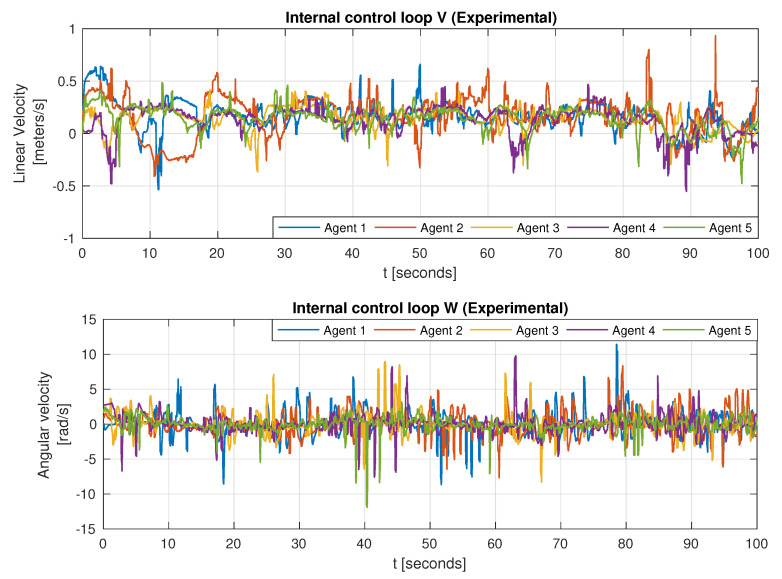
Internal control loop (experimental).

**Figure 9 entropy-25-00582-f009:**
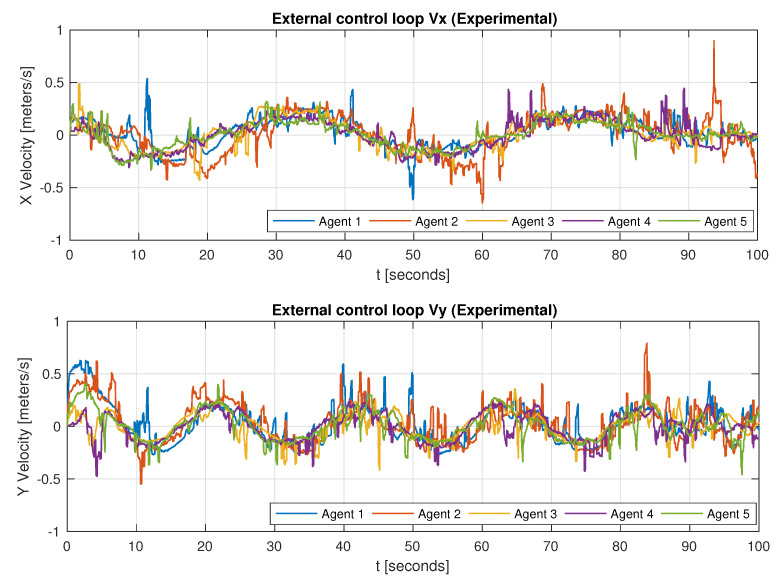
External control loop (experimental).

**Figure 10 entropy-25-00582-f010:**
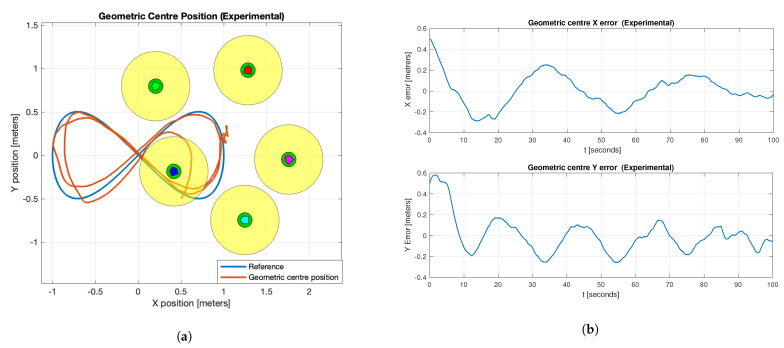
Experimental performance of the proposed controller. (**a**) Average geometric center (yellow). (**b**) Geometric center error position (experimental).

**Figure 11 entropy-25-00582-f011:**
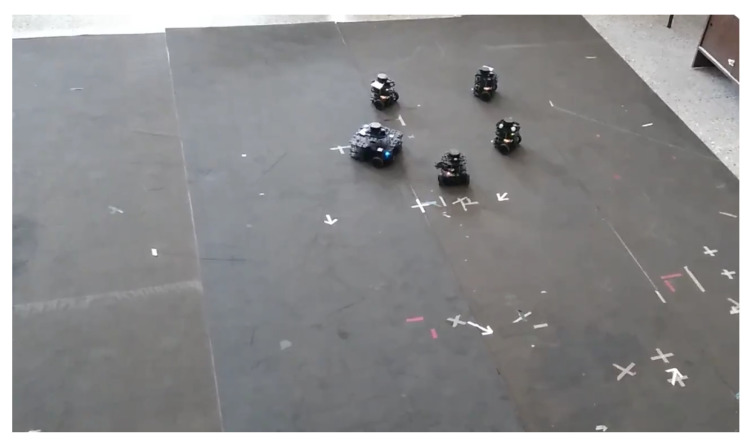
Actual experiment footage.

**Table 1 entropy-25-00582-t001:** Specifications for the desired trajectory.

a	b	d	$k_{s}$	$r_{i}$	R	$R_{d}$
1 [m]	0.5 [m]	85 [s]	0.15 [m/s]	0.4 [m]	0.6 [m]	1.0 [m]

**Table 2 entropy-25-00582-t002:** Tuning gains used on the control loop.

$k_{p}$	$k_{d}$	$k_{a}$	$k_{v}$	$k_{t}$	γ	*k*	κ	ϵ
4	4	0.5	0.995	2.3	0.49	5	13.5	0.005

**Table 3 entropy-25-00582-t003:** Initial position of each agent.

	$x_{i}$	$y_{i}$	$\theta_{i}$
Agent 1	0.5 m	−0.5 m	π/2
Agent 2	−0.5 m	−0.5 m	π/4
Agent 3	0.0 m	−0.5 m	0
Agent 4	1.0 m	−0.5 m	−π/2
Agent 5	1.5 m	−0.5 m	0

## Data Availability

The data presented in this study are available on request from the corresponding author.

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
