# Peer review of "Bio-Inspired Autonomous Navigation and Formation Controller for Differential Mobile Robots"

_entropy, 2023, doi:10.3390/e25040582_

Round 1

Reviewer 1 Report

Nice paper describing the control of autonomous fleet of robots in the plane.

I suggest to extend the state of the art section with more control strategies, for example with 'treatment of wave-based control concept' or another strategies suitable for decentralized, force-based control treatment.

According to my understanding, in equation (5), control law shoud be rather kpei + kddei than whole current equation (5), equation (5) represents equations of motion.

Comment: line 96, should ei0 rather be ei, which is used in equation (2).

I did not notice any friction model, that could help with experiment accuracy.

Reviewer 2 Report

In this article, the authors propose a decentralized controller for differential mobile robots, providing autonomous navigation and obstacle avoidance by enforcing a formation toward trajectory tracking. The control system relies on dynamic modeling, which integrates evasion forces from obstacles, formation forces, and path-following forces. The authors work is well presented, timely new and interesting. But there are some minor suggestions for further improvement of the article, such as:

1.      The grammar should be improved. Also, there are some typos throughout the article.

2.      Comparison of the proposed work with other approaches is very weak. The authors should add a proper comparison to properly validate the performance of the proposed work.

3.      The authors have added most of the figures (Figure 8-14) at the end of the paper. It would be better to put all figures at their appropriate position just after their explanation in the paper body.

4.      Please keep the text size and font style of the text in figures consistent with the text in the paper body, such as figures 3 and 4.

5.      Please add a full stop at the end of all tables and figures captions.

6.      Most of the references are old enough. There is no reference from the last 4-5 years except one reference from the year 2020.

7.      Also, the references are very limited. More references should be cited, especially from the last 2-3 years.

Reviewer 3 Report

The authors are proposing a mechanism for autonomous navigation of differential mobile robots, based on reactive, and thus biomimetic behaviors.

The paper, in general is well structured and written, while the necessary mathematic explanation of the underlying control models is more than fluent. Some minor issues should be improved though, and the following suggestions may assist:

- In section 3, it must be further justified the fact that this study analyzes mainly how to react to static obstacles rather than to dynamic ones (i.e., in lines 178-181).

- A limitation of the given approach is that the potential obstacles in the surrounding space of the reactive behavior of the robot should conform to the circular-shaped assumptions of the proposed analysis, but this is not far away from the truth from a wide variety of cases in nature, e.g., in agriculture, as the robotic solution proposed in [https://doi.org/10.3390/machines9040082].

- The authors should provide further details on the robotic vehicles (i.e., the Turtlebot3 units) being used and on the software customizations that they followed for implementing the control algorithms.

- Some Figures should be better positioned or a corresponding Appendix section to be added.

- It would be beneficial for the reader the authors to propose further fields of human activity that the mechanism being presented could be suitable for.
